# Identifying the Leading Sources of Saturated Fat and Added Sugar in U.S. Adults

**DOI:** 10.3390/nu16152474

**Published:** 2024-07-30

**Authors:** Christopher A. Taylor, Peter Madril, Rick Weiss, Cynthia A. Thomson, Genevieve F. Dunton, Michelle R. Jospe, Kelli M. Richardson, Edward J. Bedrick, Susan M. Schembre

**Affiliations:** 1Division of Medical Dietetics, The Ohio State University, Columbus, OH 43210, USA; taylor.1043@osu.edu; 2School of Health and Rehabilitative Sciences, The Ohio State University, Columbus, OH 43210, USA; madril.1@osu.edu; 3Viocare, Inc., Princeton, NJ 08542, USA; weiss@viocare.com; 4Department of Health Promotion Sciences, Mel and Enid Zuckerman College of Public Health, University of Arizona, Tucson, AZ 85724, USA; cthomson@arizona.edu; 5Department of Population and Public Health Sciences, University of Southern California, Los Angeles, CA 90039, USA; dunton@usc.edu; 6Department of Oncology, Georgetown Lombardi Comprehensive Cancer Center, Georgetown University, Washington, DC 20007, USA; michellejospe@gmail.com; 7School of Nutritional Sciences and Wellness, University of Arizona, Tucson, AZ 85721, USA; kellirichardson@arizona.edu; 8Department of Epidemiology and Biostatistics, Mel and Enid Zuckerman College of Public Health, University of Arizona, Tucson, AZ 85724, USA; edwardjbedrick@arizona.edu

**Keywords:** saturated fat, added sugars, dietary assessment, dietary surveillance, dietary intakes, ecological momentary assessment

## Abstract

The 2020–2025 Dietary Guidelines for Americans recommend limiting intakes of saturated fat and added sugars (SF/AS) to <10% total energy. Data-driven approaches to identify sources of SF/AS are needed to meet these goals. We propose using a population-based approach to identify the leading food and beverage sources of SF/AS consumed by US adults. Foods and beverages reported as consumed were assessed from two, 24 h dietary recalls (24HRDR) from 36,378 adults aged 19 years and older from the 2005–2018 National Health and Nutrition Examination Survey. Intakes of SF/AS were aggregated across both 24HRDR to identify What We Eat in America food categories accounting for ≥90% of SF/AS, respectively, by the total population and within population subgroups. Data were weighted to estimate a nationally representative sample. Ninety-five discrete food categories accounted for ≥90% of the total SF/AS intakes for >88% of the representative sample of U.S. adults. The top sources of SF were cheese, pizza, ice cream, and eggs. The leading sources of AS were soft drinks, tea, fruit drinks, and cakes and pies. This analysis reflects a parsimonious approach to reliably identify foods and beverages that contribute to SF/AS intakes in U.S. adults.

## 1. Introduction

Decreasing risks for cardiovascular disease (CVD), diabetes, hypertension, and other chronic diseases are directly anchored to overall diet quality and, more specifically, dietary components [1,2,3]. The impacts of saturated fat on human health have been widely described [1]. A meta-analysis of high-quality dietary interventions targeting reductions in dietary saturated fat intake elicited an approximately 30% reduction in cardiovascular disease [1]. Further, reduced risk of diabetes, CVD, and other causes of death has been associated with diets lower in saturated fats [1,4]. Similar deleterious impacts on cardiometabolic health have been associated with higher intakes of added sugars [5]. These behaviors have underpinned the 2020–2025 Dietary Guidelines for Americans (DGA) recommendations of limiting saturated fat and added sugar (SF/AS) intake to <10% of total daily caloric intakes [6]. Similarly, the American Heart Association has recommended dietary patterns that encourage the consumption of food sources lower in saturated fat, including plant-based and lean protein sources, while limiting foods and beverages high in added sugars [7].

The 2020–2025 Dietary Guidelines for Americans recommend that saturated fat and added sugars are limited to <10% of total daily energy intake based on their impact on type 2 diabetes risk, cardiovascular disease, and excessive body weight and obesity [6,8,9,10]. Surveillance data show mean saturated fat intake for Americans is estimated to comprise 10.7% of total daily energy intake [9]. Only 30–40% of adults consume less than 10% energy from saturated fat and there has been no decrease in mean saturated fat intake despite continued public health guidance to do so. Reductions in saturated fat intake have been shown to reduce cardiovascular events, with greater reductions in cardiovascular disease risk and cardiovascular events resulting from the replacement of saturated fat with unsaturated fats or whole grain carbohydrates [4,11]. Similarly, 70% of Americans exceed recommendations for added sugars, with 14.9% of their total daily energy intake coming from added sugars, and 6–7% of their total daily energy intake coming from sugar-sweetened beverages [8,12]. Nutrition interventions to assist adults in achieving daily targets for saturated fat and added sugar will require accurate assessment of intakes at the day-level for personalized approaches to improve diet.

Implementation of dietary behavior change interventions requires tangible strategies to improve intakes [13,14]. While specific foods are commonly targeted to address saturated fat intakes, including meats and dairy products, there are additional food sources that contribute to total dietary saturated fat intakes that are prudent to address [15,16,17]. Similarly, systematic reviews have summarized that much of the focus on added sugars has targeted sweetened beverages as well as snacks and sweets [5,18], while many other foods also contribute added sugars to overall intakes [16,19]. While these studies have extended the scope of dietary contributions to total saturated fat and added sugar intakes, a more expansive identification of key sources of SF/AS in the diet of the population (and individual) would improve content validity of targeted dietary measurements and could increase specificity of dietary behavior change interventions.

Paramount to such endeavors is the ability to identify and monitor key dietary components contributing to SF/AS intakes. Expanding assessment beyond the commonly targeted sources will allow for a more sensitive approach to address SF/AS from less prominent sources without the need to capture the full profile of foods and beverages consumed. Currently, there is a lack of validated and standardized methods to identify the primary sources of these dietary components and their proportional contribution to total daily intakes. As such, the purpose of this study was to use a data-driven approach that leverages the National Health and Nutrition Examination Survey (NHANES) dietary recall data to develop a nationally representative list of the leading food and beverage sources of SF/AS across multiple sociodemographic subgroups of adults in the U.S.

## 2. Materials and Methods

### 2.1. Study Sample

The sample for these analyses comprised U.S. adults, aged 19 years and older (*n* = 36,378), who participated in the 2005–2018 National Health and Nutrition Examination Survey (NHANES) and completed the dietary assessment. The dietary data were examined to identify the top food sources of SF/AS in the diet of American adults. NHANES is a national nutrition surveillance survey that captures health and nutritional data from non-institutionalized individuals in the U.S., using a multistage, stratified sampling methodology. Screening and consent were conducted during the NHANES in-home interview, as well as the collection of sociodemographic personal and demographic characteristics, which included age, sex, race, and ethnicity. The data collection protocols were reviewed and approved by the National Center for Health Statistics’ Research Ethics Review Board.

### 2.2. Dietary Intake Data

Two 24 h dietary recall (24HRDR) data were collected by trained interviewers using the automated multiple pass method [20]. Data from the first day of intakes were collected during the NHANES mobile examination center visit, and data from the second day of intakes were captured via telephone within 3–10 days after the initial recall. Trained interviewers recorded all foods and beverages reported as consumed between midnight and midnight on the previous day. Foods and beverages were matched to a corresponding food code from the Food and Nutrient Database for Dietary Surveys (FNDDS) [21], with the quantity and source of the food, the self-described meal by the participant, and whether it was consumed at home or away. Nutrient intake estimates from each of the foods and beverages reported were computed based on the FNDDS; MyPlate equivalents and added sugars were estimated based on the Food Patterns Equivalents Database (FPED) [22].

### 2.3. Identifying Core Foods Contributing to Saturated Fat and Added Sugars

To identify the leading food and beverage sources contributing to total SF/AS intakes, food and beverages reported during the 24HRDR were matched to the USDA What We Eat in America food categories [23] for analysis. All foods and beverages were categorized into 168 third-level food categories for the current analyses (Appendix A), as they provided the greatest specificity for discretely identifying foods and beverage sources of SF/AS. To determine the food categories contributing substantively to the population-level intakes of the SF/AS, the proportional contribution of food categories to total SF/AS intakes across both 24HRDR days was estimated using the following formula, shown for SF [24]:% of total SF from food categories = ∑(SF from food category)/∑(SF from all foods)

Estimated contributions of food and beverages to SF/AS intakes were computed for the total adult population, as well as across key sociodemographic subgroups: age (19–30 y, 31–50 y, 51–70 y, >70 y); sex; and race and ethnicity (Mexican American or other Hispanic (Hispanic or Latino), non-Hispanic White (White), non-Hispanic Black (Black), non-Hispanic Asian (Asian), Other/Multiracial). Food categories were then ranked by descending proportional contribution to total SF/AS intakes. Food categories were identified as leading sources if the cumulative contribution accounted for ≥90% of total SF/AS intakes from all food and beverages. Any food or beverage category meeting the cumulative 90% threshold for a sociodemographic subgroup was also included in the generated SF/AS food list.

To determine how the SF/AS food list assessed the amount of SF/AS in individual recalls, the sum of SF/AS intakes from foods and beverages were computed from each of the individual recalls. Within the individual foods file from the 24HRDR, foods and beverages reported by individual respondents were identified if they were from the SF/AS food list. Intakes from these food categories were aggregated on an individual basis to estimate the sum of nutrient intakes from SF/AS food lists per person per day. The percentage of intakes obtained from the SF/AS food list was computed per person, per day. Respondents were categorized into groups based on whether their individual dietary recall achieved at least 90% of their total day’s saturated fat and added sugar intakes, respectively, on 24HRDR day 1, day 2, or both days combined that were captured by the generated SF/AS food list.

### 2.4. Data Analysis

Data from the 2005–2018 NHANES were tabulated with SPSS (version 27, IBM SPSS, Armonk, NY, USA). The percentage contributions of each USDA food category to total SF/AS intakes were computed for the aggregated intakes across the day 1 and day 2 recalls. To assess the representativeness of the SF/AS food list to the total population, the proportion of the total population that received at least 90% of the days’ total intakes of SF/AS were computed. These proportions were also estimated across sociodemographic subgroups. As supplemental analyses, proportional contributions to total intakes were computed for all adults, and across sex, age, and race and ethnicity. Mean (and standard errors) of the intakes from each of the 24HRDR, and the average of the two 24HRDR were computed for the total day’s nutrient intakes, the nutrient intakes from the SF/AS food list, and the percentage of the days’ total intakes. All analyses were conducted using SPSS Complex Samples (version 27, IBM SPSS, Armonk, NY, USA). Population dietary data sampling weights were applied to account for the complex sampling design and create nationally representative estimates.

## 3. Results

Dietary intake data were available from 36,378 adults, with representation from the following subgroups: males (*n* = 17,701); females (*n* = 18,677); 19–30 year olds (*n* = 7715); 31–50 year olds (*n* = 11,871); 51–70 year olds (*n* = 11,386); >70 year olds (*n* = 5406); Mexican American (*n* = 5805); other Hispanic (*n* = 3452); non-Hispanic White (*n* = 15,363); non-Hispanic Black (*n* = 7953); non-Hispanic Asian (*n* = 2328); Other or Multiracial (*n* = 1477). After analysis, dietary intakes from The What We Eat in America food categories that accounted for the most SF/AS across the total population, age, sex, ethnicity, and race are presented in Table 1 and Table 2, respectively.

### 3.1. Food Sources of Saturated Fat

Of the total 168 categories, 54 food categories contributed to 90% of the total saturated fat intakes, and 71 food categories contributed to 90% of the total saturated fat intakes across any of the age, sex, or race and ethnicity subgroups (Table 1). Cheese, pizza, ice cream, eggs and omelets, burritos and tacos, and chicken were the leading sources of saturated fat, accounting for 25% of total saturated fat intakes. In adults over 70 years old, ice cream was the top source of saturated fat, with comparatively lower contributions from pizza, burritos and tacos, and burgers. Older adults also had a greater contribution to intakes from butter and animal fats, and baked goods (cakes and pies, as well as cookies and brownies). When stratified by race and ethnicity, Hispanic or Latinos had the greatest contribution to saturated fat from burritos and tacos, non-Hispanic Blacks had the greatest contribution to saturated fat from chicken, and Asians had the greatest contribution to saturated fat from nuts and seeds.

### 3.2. Food Sources of Added Sugars

Out of 168 total categories, 30 food categories accounted for 90% of the added sugars in the total population, with 49 food categories accounting for 90% of added sugar intakes across any of the age, sex, or race and ethnicity subgroups (Table 2). Nearly half of the intakes of added sugars from adults were from five sources: soft drinks, tea, fruit drinks, cakes and pies, and sugar and honey. The top sources of added sugars for persons over 70 years were ice cream and frozen dairy desserts, cookies and brownies, and jams, syrups, and toppings. Added sugar from soft drinks was highest among Hispanics or Latinos as compared to other races and ethnicities. Among Asians, soft drinks, followed by sugars and honey, cookies and brownies, tea, and cakes and pies, were the main sources of AS.

### 3.3. Contributions of Food Sources of Saturated Fat and Added Sugars to Total Intakes

Mean intakes of nutrients from the total day, mean intakes contributed by the SF/AS food list, and the percent of the day’s intakes obtained from the SF/AS food list are presented in Table 3. Across the two days of 24HRDR, a substantive mean portion of total SF/AS intakes (>94%, respectively) was obtained from the derived SF/AS food list. The SF/AS food list was successful at accounting for at least 90% of SF/AS intakes from the given day for over 88% of adults. For the other macronutrients, the contribution of the foods and beverages in the SF/AS food list accounted for intakes of protein (≥91%), carbohydrates (≥82%), and total fat (≥95%) relative to the total day; however, the percentage of adults obtaining at least 90% of their day’s intakes from the food list was only greater than 90% for saturated fat and added sugars. The proportions of adults that obtained at least 90% of their total day’s intakes for energy, protein, carbohydrate, and total fat from the SF/AS food list were considerably lower. These patterns were evident across day 1, day 2, and the average of both days.

Table 4 shows the proportion of adults consuming each food category across the days of intake. Across both days, coffee, yeast breads, and cheese intakes were reported as the most regularly consumed foods on days 1 and 2, with ≥52% reported as consumed on one or both days; however, 41.1 to 47.9% reported these foods as not consumed on either day. More than 30% of adults reported consuming soft drinks, tea, eggs and omelets, chicken (whole pieces), other vegetables and combinations, tomato-based condiments, cookies and brownies, salad dressings and vegetable oils, and cold cuts and cured meats on either of the two days. Cream and cream substitutes (contributing both saturated fat and added sugar) were reported as being consumed on both days by 15.3% and on any day by 27.5% of adults. Pizza (the second leading food category contributing to total saturated fat intakes; Table 1) was reported as being eaten by 16.4% on at least one day and 1.3% on both days.

### 3.4. Representativeness of the SF/AS Food List by Sociodemographic Characteristics

The percentages of adults who obtained at least 90% of their day’s intakes of SF/AS from the derived food list by sociodemographic characteristics are presented in Table 5. The generated SF/AS food list captured ≥90% of saturated fat and added sugar intakes from 24HRDR day 1 and day 2 for a proportionally greater number of males (92% and 89%, respectively), adults who were 19–50 years old (91% and 89%, respectively), and Mexican American adults (93% and 92%, respectively). The derived SF/AS food list captured ≥90% of SF/AS intakes from 24HRDR day 1 and day 2 for proportionally fewer Asian adults (85% and 86%, respectively).

## 4. Discussion

Central to dietary interventions is the identification of key dietary targets that could be modified to impact cardiometabolic health and the ability to develop actionable strategies for behavior change [13,14]. This can be achieved at both the population and individual level with increasing degrees of personalization. Efforts to modify the predominant sources of SF/AS intakes in the American diet have demonstrated success [25,26]; however, overly generalized interventions focused on only a few major sources of AS/SF, which may or may not be consumed by certain individuals or subgroups of individuals, may hinder the effectiveness of large-scale dietary interventions. Additionally, assessing dietary intakes that are efficient and specific to the target behaviors is also of concern. A more nuanced conceptualization of the dietary sources of SF/AS in a diverse food culture, such as that in the U.S. [17], creates an opportunity for more robust dietary interventions while facilitating a more sensitive assessment of total SF/AS intakes.

Considerable nutrition science literature has focused on reducing the consumption of dairy [27], red and processed meats [28], and sugar-sweetened beverages [29,30] as a focus of reductions in saturated fats and added sugars. Consistent with the evidence base, the 2020–2025 Dietary Guidelines for Americans recommend that saturated fat and added sugars are limited to <10% of total daily energy intake [6,8,9,10]. While these foods and beverages contribute greatest to these intakes [17,19], our data demonstrate a wider variety of foods contributing to total SF/AS intakes, with notable differences in food sources based on sociodemographic characteristics. The high threshold used in the current analyses (≥90%) illuminates the foods and beverages with lesser quantities of SF/AS that are consumed in greater quantities to elucidate their meaningful, overall contributions to SF/AS intakes. For example, among older adults, ice cream contributes a greater amount to SF/AS intakes than is demonstrated for other age groups. Non-Hispanic Black and non-Hispanic Asian groups consumed considerably more saturated fat from fish and seafood than other groups. Furthermore, we revealed considerable variability in the consumption of foods and beverages across the days of 24HRDR than previously noted [16,17,19]. As such, focusing on some of the top sources of SF/AS may be less effective on a given day, based on interindividual variability in consumption patterns. Identifying more food and beverage sources of SF/AS intakes increases the likelihood of capturing the diversity of dietary intakes and helps account for the normal episodic consumption patterns of several of the identified sources of SF/AS while also capturing the variability in sources of key subgroups [17].

While addressing these dietary intakes is critical to dietary interventions that aim to promote chronic disease risk reduction, it is equally important to reliably quantify changes in intakes—a long-standing challenge to dietary surveillance [31,32,33]. The complexity of dietary intake assessment represents a balance of respondent burden, length of time being evaluated, parsimony to the key nutrients of interest, and the precision of the resulting estimates produced [34]. The antithesis of accuracy is ambiguity. Interviewer-assisted 24HRDR, the gold standard of dietary assessment [35], captures the full cadre of foods reported for a single day to elicit a more precise estimate of nutrient intakes [20,36]. However, these assessments that accurately estimate all nutrients are beyond the scope of this focus on saturated fat and added sugars, and the respondent burden of 24HRDR remains high [37]. Conversely, food frequency questionnaires and dietary screeners offer inspection of intake patterns that better address day-to-day variability at the known expense of the precision of daily intake estimates [34,37]. Researchers must balance the decisions for precision of daily intakes to the intraindividual variability to assess patterns of behavior when evaluating the impact of dietary interventions [31]. These efforts to identify the leading sources of SF/AS are an example of creating parsimony in assessment while reducing respondent burden.

The implications of identifying the top sources of SF/AS in the diverse American diet extend to the development of more targeted, lower burden, and more sensitive dietary assessment and (personalized) intervention strategies consistent with practice guidelines set forth by the Academy of Nutrition and Dietetics [4]. The list generated from these analyses represents 95 discrete food and beverage sources, and the items have been clustered into 12 subgroups for more rapid identification and less respondent burden. This enhances precision of estimation of subgroups while limiting the initial response options in dietary assessment. One such strategy being explored is ecological momentary diet assessment, whereby respondents are asked to identify, from a parsimonious list of targeted foods and beverages (which, here, contribute to SF/AS intakes), the foods and beverages they recently consumed [38]. The translation of these food lists to a user-friendly mobile platform has further implications for the modernization of dietary surveillance and personalized dietary interventions [39].

There are several factors that should be considered when interpreting these data. Utilizing dietary intake assessments from the What We Eat in America data from NHANES offers a large, nationally representative sample of adults from whom dietary intakes were collected. The data were collected via the validated automated multiple-pass method by trained interviewers to increase validity of the recalls [35], but there may be differences in estimates based on in-person vs. telephone-based interviews. Additionally, the limited number of dietary recalls cannot be assumed to represent usual intakes for respondents; thus, these data represent the distribution of intakes on the days of intakes. Utilization of the second day of intakes created an opportunity for a validation of the food list created on the first day of intake, with a high degree of congruence across days of recalls. Furthermore, the analyses conducted by race and ethnicity were limited to oversampled subgroups and do not represent other subgroups or fully capture the diversity of intakes within sampled groups.

## 5. Conclusions

The identification of the leading food and beverage sources of SF/AS is critical for the development of more targeted dietary surveillance and intervention efforts to promote cardiometabolic health at the population and individual levels. This analysis derived a list of 95 unique food and beverages categories that contributed ≥90% of the total SF/AS intakes of a sociodemographic diverse population of U.S. adults. These data report a greater diversity of foods that contribute to total saturated fat and added sugars than previously explored in the literature. This broader assessment of food sources can be leveraged to develop novel dietary surveillance and intervention tools. The methods by which we performed our analysis can be replicated for other nutrients of interest. Recognizing the broader contributors of SF/AS to the diet provides nutrition education professionals with greater capacity to support individuals in achieving the US Dietary Guidelines for Americans.

## Figures and Tables

**Table 1 nutrients-16-02474-t001:** Top 25 USDA What We Eat in America food categories contributing to saturated fat intakes in US adults by age, sex, and race and ethnicity.

**Food Category**	**19–30 y**	**31–50 y**	**51–70 y**	**>70 y**	**Male**	**Female**	**Total**
Cheese	8.8%	8.5%	8.1%	6.7%	8.1%	8.5%	8.3%
Pizza	7.8%	5.6%	3.7%	2.2%	6.0%	4.3%	5.2%
Ice cream and frozen dairy desserts	3.3%	3.7%	5.1%	7.9%	4.3%	4.6%	4.4%
Eggs and omelets	3.7%	3.7%	3.9%	4.1%	3.8%	3.8%	3.8%
Burritos and tacos	5.0%	4.1%	2.4%	1.2%	4.0%	2.9%	3.5%
Chicken, whole pieces	3.2%	2.9%	2.4%	1.8%	2.9%	2.5%	2.7%
Butter and animal fats	1.4%	2.3%	3.5%	4.6%	2.4%	3.0%	2.7%
Burgers (single code)	3.7%	2.7%	2.1%	1.2%	3.1%	2.0%	2.6%
Cakes and pies	1.8%	2.5%	2.9%	3.7%	2.3%	2.9%	2.6%
Nuts and seeds	1.6%	2.3%	3.3%	2.7%	2.5%	2.5%	2.5%
Candy containing chocolate	1.9%	2.5%	2.8%	2.4%	2.0%	3.0%	2.4%
Cookies and brownies	2.2%	2.1%	2.5%	3.5%	2.3%	2.5%	2.4%
Meat mixed dishes	1.9%	2.1%	2.6%	2.9%	2.4%	2.2%	2.3%
Milk, reduced fat	2.2%	2.0%	2.1%	3.0%	2.3%	2.1%	2.2%
Beef, excludes ground	2.0%	2.3%	2.0%	1.6%	2.4%	1.6%	2.1%
Doughnuts, sweet rolls, pastries	1.8%	2.0%	2.1%	2.3%	2.0%	1.9%	2.0%
Milk, whole	2.2%	1.9%	1.8%	2.0%	2.0%	1.9%	2.0%
Pasta mixed dishes, excl. mac and cheese	2.4%	1.9%	1.7%	1.8%	1.8%	2.1%	1.9%
Sausages	1.4%	1.8%	1.8%	1.8%	2.0%	1.4%	1.7%
Cold cuts and cured meats	1.4%	1.6%	1.9%	2.0%	1.9%	1.4%	1.7%
Other Mexican mixed dishes	2.2%	1.8%	1.1%	0.5%	1.5%	1.6%	1.6%
Cream and cream substitutes	0.8%	1.7%	1.9%	1.7%	1.3%	1.9%	1.6%
Salad dressings and vegetable oils	1.4%	1.5%	1.6%	1.6%	1.3%	1.8%	1.5%
Ground beef	1.7%	1.5%	1.5%	1.1%	1.7%	1.2%	1.5%
French fries and other fried white potatoes	1.9%	1.6%	1.2%	0.8%	1.6%	1.3%	1.5%
**Food Category**	**Mexican American**	**Other Hispanic**	**Non-Hispanic White**		**Non-Hispanic Black**	**Non-Hispanic Asian**	**Other/** **Multiracial**
Cheese	5.3%	6.9%	8.4%		5.1%	3.5%	7.1%
Pizza	4.3%	5.5%	5.2%		5.4%	4.3%	5.0%
Ice cream and frozen dairy desserts	2.5%	4.1%	4.7%		3.7%	3.3%	3.8%
Eggs and omelets	4.9%	5.1%	3.8%		4.1%	4.3%	3.8%
Burritos and tacos	12.4%	4.5%	4.1%		2.6%	2.5%	4.2%
Chicken, whole pieces	2.4%	3.4%	1.8%		6.4%	3.7%	2.7%
Butter and animal fats	1.1%	1.7%	3.0%		1.4%	1.6%	2.1%
Burgers (single code)	3.2%	3.1%	2.8%		4.4%	1.8%	4.2%
Cakes and pies	1.9%	2.4%	2.6%		3.1%	2.4%	2.9%
Nuts and seeds	1.6%	1.7%	2.9%		1.9%	4.4%	2.5%
Candy containing chocolate	1.2%	1.4%	2.5%		2.1%	2.1%	2.0%
Cookies and brownies	2.6%	2.3%	2.8%		2.9%	2.9%	3.0%
Meat mixed dishes	1.7%	1.6%	2.4%		1.7%	2.2%	2.2%
Milk, reduced fat	2.4%	2.0%	1.9%		1.1%	2.1%	2.2%
Beef, excludes ground	2.4%	2.4%	1.7%		1.9%	1.7%	1.9%
Doughnuts, sweet rolls, pastries	2.7%	2.2%	2.3%		2.2%	2.1%	2.5%
Milk, whole	2.0%	1.9%	1.8%		1.9%	2.6%	1.5%
Pasta mixed dishes, excl. mac and cheese	1.4%	2.0%	2.1%		2.4%	1.7%	2.6%
Sausages	1.1%	1.4%	1.6%		2.4%	0.9%	1.5%
Cold cuts and cured meats	0.8%	1.5%	1.7%		1.4%	1.1%	1.6%
Other Mexican mixed dishes	7.2%	3.3%	1.3%		0.5%	0.8%	1.8%
Cream and cream substitutes	1.3%	1.2%	1.7%		1.0%	1.2%	1.4%
Salad dressings and vegetable oils	0.9%	1.1%	1.5%		1.6%	1.1%	1.4%
Ground beef	0.8%	1.0%	1.2%		0.9%	0.6%	1.3%
French fries and other fried white potatoes	1.2%	1.1%	1.2%		1.9%	1.1%	1.5%

**Table 2 nutrients-16-02474-t002:** Top 25 USDA What We Eat in America food categories contributing to added sugar intakes in US adults by age, sex, and race and ethnicity.

**Food Category**	**19–30 y**	**31–50 y**	**51–70 y**	**>70 y**	**Male**	**Female**	**Total**
Soft drinks	33.6%	28.9%	19.3%	11.3%	28.8%	22.5%	26.0%
Tea	6.4%	7.0%	7.5%	5.5%	6.6%	7.3%	6.9%
Fruit drinks	8.8%	6.6%	5.7%	5.1%	6.8%	6.8%	6.8%
Cakes and pies	3.9%	6.0%	7.5%	10.8%	5.7%	7.0%	6.3%
Sugars and honey	3.3%	5.4%	6.1%	5.2%	4.9%	5.2%	5.0%
Ice cream and frozen dairy desserts	3.3%	3.8%	5.8%	9.3%	4.5%	4.9%	4.7%
Cookies and brownies	3.6%	3.9%	5.2%	7.6%	4.2%	4.8%	4.5%
Candy containing chocolate	2.4%	3.2%	4.1%	3.6%	2.8%	3.9%	3.3%
Sport and energy drinks	4.5%	3.3%	1.4%	0.6%	4.0%	1.3%	2.8%
Ready-to-eat cereal, higher sugar (>21.2 g/100 g)	3.2%	2.4%	2.4%	3.2%	2.6%	2.8%	2.7%
Jams, syrups, toppings	2.0%	2.4%	3.1%	4.0%	2.7%	2.5%	2.6%
Candy not containing chocolate	2.2%	2.2%	2.8%	1.8%	2.0%	2.7%	2.3%
Doughnuts, sweet rolls, pastries	1.9%	2.2%	2.4%	2.6%	2.3%	2.1%	2.2%
Yeast breads	1.1%	1.3%	1.9%	2.6%	1.5%	1.6%	1.5%
Cream and cream substitutes	0.8%	1.6%	1.9%	1.4%	1.1%	1.9%	1.4%
Biscuits, muffins, quick breads	1.1%	1.1%	1.6%	2.5%	1.2%	1.6%	1.4%
Liquor and cocktails	1.1%	1.0%	1.1%	0.7%	1.0%	1.2%	1.0%
Yogurt, regular	0.8%	0.8%	1.2%	1.3%	0.6%	1.3%	0.9%
Tomato-based condiments	1.0%	0.9%	0.9%	0.6%	1.1%	0.7%	0.9%
Ready-to-eat cereal, lower sugar (≤21.2 g/100 g)	0.7%	0.7%	1.0%	1.6%	0.8%	0.9%	0.9%
Coffee	1.1%	0.9%	0.6%	*	0.6%	1.1%	0.8%
Salad dressings and vegetable oils	0.5%	0.7%	1.0%	1.0%	0.7%	0.9%	0.8%
Cereal bars	0.7%	0.8%	0.7%	0.5%	0.6%	0.8%	0.7%
Rolls and buns	0.7%	0.7%	0.8%	0.8%	0.8%	0.6%	0.7%
Soft drinks	33.6%	28.9%	19.3%	11.3%	28.8%	22.5%	26.0%
**Food Category**	**Mexican American**	**Other** **Hispanic**	**Non-** **Hispanic White**		**Non-Hispanic Black**	**Non-Hispanic Asian**	**Other/** **Multiracial**
Soft drinks	33.6%	27.8%	22.6%		25.3%	15.7%	26.8%
Tea	5.4%	6.2%	8.7%		8.2%	6.1%	11.2%
Fruit drinks	7.6%	9.2%	3.8%		12.0%	5.2%	6.3%
Cakes and pies	4.3%	5.3%	5.9%		5.9%	6.0%	5.6%
Sugars and honey	5.4%	7.9%	4.7%		5.5%	9.0%	5.9%
Ice cream and frozen dairy desserts	2.8%	4.1%	5.2%		3.4%	4.3%	3.0%
Cookies and brownies	4.5%	4.0%	5.1%		4.4%	6.3%	4.7%
Candy containing chocolate	1.8%	1.8%	3.6%		2.6%	3.3%	2.2%
Sport and energy drinks	4.3%	2.7%	3.2%		2.8%	2.1%	3.5%
Ready-to-eat cereal, higher sugar (>21.2 g/100 g)	2.4%	1.8%	2.8%		2.7%	2.1%	2.7%
Jams, syrups, toppings	1.3%	2.1%	2.9%		2.3%	2.2%	1.7%
Candy not containing chocolate	1.6%	1.2%	2.5%		3.1%	2.2%	2.4%
Doughnuts, sweet rolls, pastries	3.4%	2.2%	2.2%		1.7%	2.7%	2.0%
Yeast breads	1.2%	1.4%	1.5%		1.2%	2.0%	1.3%
Cream and cream substitutes	1.5%	1.3%	1.8%		1.1%	1.4%	2.1%
Biscuits, muffins, quick breads	0.6%	1.0%	1.5%		1.2%	1.8%	0.8%
Liquor and cocktails	0.9%	0.9%	1.5%		1.2%	1.1%	0.9%
Yogurt, regular	0.8%	1.1%	0.9%		*	1.0%	*
Tomato-based condiments	0.7%	0.8%	1.0%		1.0%	0.8%	0.8%
Ready-to-eat cereal, lower sugar (≤21.2 g/100 g)	0.9%	0.6%	1.0%		*	0.8%	0.5%
Coffee	1.0%	1.5%	0.9%		0.5%	2.3%	1.7%
Salad dressings and vegetable oils	*	*	0.9%		0.6%	0.7%	0.5%
Cereal bars	0.5%	0.7%	0.9%		0.5%	0.7%	0.5%
Rolls and buns	*	0.5%	0.7%		0.4%	0.5%	0.5%
Soft drinks	33.6%	27.8%	22.6%		25.3%	15.7%	26.8%

* Indicates the food or beverage category did not contribute to leading sources of added sugars in the population or subpopulation.

**Table 3 nutrients-16-02474-t003:** Mean contribution of intakes from the SF/AS food list to total day’s intakes and the proportion of U.S. adults obtaining ≥90% of the day’s intakes from the food list (*n* = 36,355).

Day	Dietary Component	Intakes from SF/AS Food List ^1^	Intakes from Total Day ^1^	% of Day from SF/AS Food List ^1^	Percent Obtaining ≥90% of Day’s Intakes from SF/AS Food List ^2^
Day 1 *n* = 36,355	Energy (kcal)	1888 (8)	2151 (8)	87.3% (0.1)	52.3%
	Protein (gm)	75.8 (0.4)	83 (0.4)	91.0% (0.1)	71.5%
	Carbohydrate (gm)	215 (1)	256 (1)	83.0% (0.2)	42.4%
	Total fat (gm)	80.1 (0.4)	83.2 (0.4)	95.6% (0.1)	87.3%
	Saturated fat (gm)	26.5 (0.2)	27.2 (0.2)	96.5% (0.1)	90.3%
	Added sugars (tsp. eq.)	17.3 (0.2)	17.8 (0.2)	95.0% (0.1)	88.2%
Day 2 *n* = 31,756	Energy (kcal)	1780 (9)	2030 (10)	86.9% (0.2)	51.1%
	Protein (gm)	74.1 (0.4)	81.2 (0.4)	90.5% (0.1)	70.7%
	Carbohydrate (gm)	203 (1)	244 (1)	81.8% (0.2)	39.9%
	Total fat (gm)	75.3 (0.5)	78.3 (0.5)	95.3% (0.1)	86.9%
	Saturated fat (gm)	24.9 (0.2)	25.6 (0.2)	96.1% (0.1)	89.5%
	Added sugars (tsp. eq.)	15.5 (0.2)	15.9 (0.2)	94.7% (0.2)	88.4%
Average *n* = 36,355	Energy (kcal)	1837 (8)	2097 (8)	87.3% (0.1)	48.5%
	Protein (gm)	75 (0.3)	82.3 (0.3)	91.0% (0.1)	69.3%
	Carbohydrate (gm)	209 (1)	250 (1)	82.6% (0.2)	35.7%
	Total fat (gm)	77.9 (0.4)	81 (0.4)	95.7% (0.1)	87.5%
	Saturated fat (gm)	25.8 (0.1)	26.5 (0.1)	96.7% (0.1)	91.4%
	Added sugars (tsp. eq.)	16.5 (0.2)	17 (0.2)	95.8% (0.1)	88.2%

^1^ Data presented as mean (SE); ^2^ Reflects the percent of the sample that obtained ≥90% of their intake of each nutrient for the total days from the food in the SF/AS food list.

**Table 4 nutrients-16-02474-t004:** Percent of the sample reporting the top 25 foods or beverage categories on the day of record.

Food Category			Both Days			
	Reported on Day 1	Reported on Day 2	Not Reported on Either Day	Reported on 1 Day	Reported on Both Days	Reported on Either Day
Soft drinks	53.1%	53.5%	41.1%	16.8%	42.1%	58.9%
Tea	39.3%	41.8%	43.5%	36.3%	20.3%	56.5%
Fruit drinks	36.9%	34.6%	47.9%	36.6%	15.5%	52.1%
Cakes and pies	30.9%	27.2%	61.7%	22.1%	16.2%	38.3%
Sugars and honey	27.1%	26.9%	64.3%	20.6%	15.1%	35.7%
Ice cream and frozen dairy desserts	21.2%	23.5%	66.1%	26.2%	7.8%	33.9%
Cookies and brownies	20.1%	21.6%	66.1%	28.5%	5.4%	33.9%
Candy containing chocolate	20.9%	21.8%	66.7%	26.1%	7.2%	33.3%
Sport and energy drinks	22.2%	18.7%	67.0%	27.6%	5.4%	33.0%
Ready-to-eat cereal, higher sugar (>21.2 g/100 g)	21.8%	19.2%	67.7%	25.6%	6.7%	32.3%
Jams, syrups, toppings	20.0%	20.6%	68.1%	25.4%	6.6%	31.9%
Candy not containing chocolate	19.4%	19.8%	68.7%	25.4%	5.9%	31.3%
Doughnuts, sweet rolls, pastries	23.4%	22.1%	70.2%	16.8%	13.0%	29.8%
Yeast breads	19.8%	19.2%	71.1%	20.7%	8.2%	28.9%
Cream and cream substitutes	22.8%	22.4%	72.5%	12.3%	15.3%	27.5%
Biscuits, muffins, quick breads	17.4%	16.3%	72.5%	23.1%	4.3%	27.5%
Liquor and cocktails	16.8%	15.4%	73.5%	22.7%	3.8%	26.5%
Yogurt, regular	16.2%	14.4%	74.5%	22.1%	3.3%	25.5%
Tomato-based condiments	14.8%	15.0%	76.1%	19.8%	4.1%	23.9%
Ready-to-eat cereal, lower sugar (≤21.2 g/100 g)	15.1%	14.0%	76.8%	18.7%	4.4%	23.2%
Coffee	15.4%	15.9%	78.6%	13.2%	8.1%	21.4%
Salad dressings and vegetable oils	14.7%	11.9%	78.7%	17.2%	4.1%	21.3%
Cereal bars	12.6%	12.7%	79.1%	18.0%	3.0%	20.9%
Rolls and buns	13.1%	11.8%	79.8%	16.7%	3.5%	20.2%
Soft drinks	12.3%	11.6%	79.9%	17.7%	2.4%	20.1%

**Table 5 nutrients-16-02474-t005:** Percent by population subgroups with at least 90% of the day’s intakes of saturated fat and added sugars obtained from the food list.

Characteristic	Category	Met ≥90% of Saturated Fat from SF/AS Food List			Met ≥90% of Added Sugars from SF/AS Food List		
		Day 1	Day 2	Total	Day 1	Day 2	Total
Gender	Male	91.3%	90.6%	92.5%	89.0%	89.1%	89.4%
	Female	89.4%	88.5%	90.4%	87.5%	87.7%	87.0%
Age categories	19–30 years	91.9%	90.3%	92.9%	90.0%	88.2%	89.7%
	31–50 years	90.3%	90.0%	91.7%	88.7%	89.2%	88.7%
	51–70 years	89.9%	89.3%	90.8%	87.0%	88.0%	87.3%
	>70 years	88.3%	87.1%	88.9%	86.6%	87.3%	85.8%
Race/Ethnicity	Mexican American	92.9%	92.5%	93.5%	92.1%	91.3%	91.8%
	Other Hispanic	90.1%	88.8%	91.5%	89.7%	89.0%	89.2%
	Non-Hispanic White	90.4%	89.5%	91.7%	87.0%	87.6%	86.9%
	Non-Hispanic Black	89.2%	88.4%	90.0%	91.6%	90.9%	91.7%
	Asian	85.7%	86.1%	85.8%	86.2%	85.3%	86.4%
	Other or Multiracial	90.1%	90.3%	90.6%	90.1%	89.3%	90.6%

## Data Availability

The data presented in this study were derived from the following resources available in the public domain: Demographics: https://wwwn.cdc.gov/nchs/nhanes/search/datapage.aspx?Component=Demographics (accessed on 5 October 2021); Individual Foods Files: https://wwwn.cdc.gov/nchs/nhanes/search/datapage.aspx?Component=Dietary (accessed on 5 October 2021); Food Patterns Equivalents Database: https://www.ars.usda.gov/northeast-area/beltsville-md-bhnrc/beltsville-human-nutrition-research-center/food-surveys-research-group/docs/fped-methodology/ (accessed on 5 October 2021); What We Eat in America Food Categories: https://www.ars.usda.gov/northeast-area/beltsville-md-bhnrc/beltsville-human-nutrition-research-center/food-surveys-research-group/docs/dmr-food-categories/ (accessed on 5 October 2021).

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
