# Peer review of "Identifying the Leading Sources of Saturated Fat and Added Sugar in U.S. Adults"

_nutrients, 2024, doi:10.3390/nu16152474_

Round 1
Reviewer 1 Report (New Reviewer)
Comments and Suggestions for Authors
I am very grateful to you for the invitation to review the manuscript nutrients-3096264 by Taylor and coauthors "Identifying the Leading Sources of Saturated Fat and Added Sugar in U.S. Adults”. This research aimed to use a data-driven approach that leverages the National Health and Nutrition Examination Survey (NHANES) dietary recall data to develop a nationally representative list of the leading food and beverage sources of SF/AS across multiple sociodemographic subgroups of adults in the U.S. The work is interesting but needs several adjustments to increase the quality of the material.
Comments:
- Lines 18-19: Better specify the objective of the work.
- Abstract: Based on the indicators, what measures or suggestions do the authors have?
- Abstract: The conclusion should align with the objectives of the work and indicate perspectives to mitigate this problem.
- Lines 30-31: Change the repeated keywords to different terms not included in the title.
- Lines 34-36: Consider using only the necessary references for the sentence. There is an excess of references for a generic sentence.
- Lines 36-37: Consider using only the necessary references for the sentence. There is an excess of references for a generic sentence. Seven references for a generic statement are inconsistent.
- Introduction: The authors should deepen the discussion on the effect or impact of SF/AS on the body.
- Lines 59-65: It is not clear what measures can be taken or suggestions beyond identification.
- Line 72: Better specify the term “NHANES” in the first instance.
- Line 82: Better explain the 24HRDR.
- Materials and Methods: Better specify how the profile of the product components was reported, given the diversity of formulations on the market.
- Lines 222-223: Strategies for promoting healthy eating and health should be addressed.
- Lines 225-226: The authors need to better specify the measures taken so far aimed at this point.
- Discussion: Include public policies implemented to improve adequate access to food and reduce harmful components.
- Discussion: Deepen the discussion on possible measures and perspectives based on the presented and discussed results.
- The supplementary material indicated in the text and mentioned in the item is not available for analysis.
Author Response
See attached response file

Reviewer 2 Report (New Reviewer)
Comments and Suggestions for Authors
The study's objective was to develop a representative list of the main sources of foods and beverages of saturated fats and added sugars across multiple sociodemographic subgroups of adults in the US. The study is scientifically significant and provides various pertinent insights to the field. However, it is crucial to review certain points to ensure higher levels of excellence for publication.
Consider replacing the keywords in the abstract that are identical to those in the title to expand the study's reach in databases. It is also recommended to define what "SF/AS" (saturated fat/added sugars) means the first time it appears for clarity.
The introduction mentions several dietary recommendations but does not sufficiently explore the reasons behind them. Including brief explanations or examples of the negative impacts of saturated fats and added sugars on health could be beneficial.
What was the specific criterion for selecting the age range of 19 years and older? Is there a reason for not including adolescents or institutionalized elderly individuals?
Consider describing why the years 2005-2018 were chosen and if there are any limitations associated with this time period. Were the data weighted to reflect the US population? If so, how were these weightings applied?
How was the consistency and accuracy of the telephone interviews compared to those conducted at the mobile examination center ensured? How were potential memory lapses or underreporting in the 24HRDRs handled? Was any statistical adjustment applied to mitigate these biases? Providing data or statistics quantifying this congruence would strengthen the validity of the conclusions. Additionally, the meaning of the abbreviation 24HRDR should be presented the first time it is used in the text.
The methodology of the "automated multiple pass method" could be briefly explained for better understanding, even with the reference. Furthermore, is this methodology validated specifically for this population? Are there studies supporting its accuracy and reliability in this context?
The "Dietary Intake Data" section mentions "MyPlate equivalents" and "Food Patterns Equivalents Database (FPED)" without explanation. Consider briefly explaining what they are and how they are used.
The formula "% of total SF from food categories = ∑(SF from food category)/ ∑(SF from all foods)" should be formatted consistently and clearly to ensure readers can easily understand it. Additionally, what was the justification for using this specific formula to calculate the proportional contribution of each food category? Is this formula widely accepted in the literature?
Regarding the sociodemographic subgroups, was there control for possible confounding variables within these subgroups, such as physical activity levels or other health conditions? Is there a justification for the demographic distribution of the participants? How does this distribution influence the generalization of the results to the total population?
Authors should explore seasonal or regional variation across states in the consumption of specific foods that may influence SF/AS intake.
Regarding Tables 1 and 2, the formatting needs to be revised, as there is spacing in the middle suggesting that these are separate tables.
Finally, how does this study contribute to the existing literature on sources of SF/AS in the American diet? Are there any conflicting or confirmatory findings concerning previous studies?
Author Response
See attached response file

Reviewer 3 Report (New Reviewer)
Comments and Suggestions for Authors
This paper presented the findings of the leading saturated fat and added sugar sources in the consumed foods of U.S. adults. The results were numerous and obtained from a 14-year national health and nutrition examination survey, which were used to identify the top sources of saturated fat and added sugar. The data were helpful for the recognition of the broader contributors of saturated fat and added sugar to the diet providing nutrition for Americans. The content was acceptable and the analysis was reasonable. I suggest that this paper be revised in the following aspects:
1. The introduction part should present the background, necessity and importance of the study. But the authors failed to show the above content. Some revisions should be conducted. For example, the existed findings related to the current study can be simply described. The necessity and importance of the current study can be explained in detail.
2. Tables may be one general presentation format of the detailed data, however, figures can show the findings more clearly and visually. Please consider for the replacement of some tables by figures.
3. Is the statistically analysis suitable for the data analysis in the current study?
4. Reference 23 “What We Eat In American Food Categories” is a book? Journal article? Website paper? Or others? Please clarify it.
5. The conclusion part should be expanded and more detailed findings can be shown in this section.
Author Response
See attached response file

Round 2
Reviewer 1 Report (New Reviewer)
Comments and Suggestions for Authors
Dear Editor,
There are many sentences with generic information and an excess of references. Several of the suggestions were not incorporated into the text. Therefore, I recommend including the required information.
Author Response
Response: We thank the reviewer for providing clarification on their original critiques and describe below the additional revisions that have been made to the revised manuscript in response to the points below.
In the first round of review, the authors were asked to report on current government policies or goals regarding fat and sugar consumption to mitigate the negative impact on public health (at least in the U.S.). However, the authors merely stated that this was beyond the scope of the work. In addition to presenting results, I believe that a scientific paper should outline the current scenarios within the topic and suggest directions for overcoming the described issues.
Response: We have reported on the current government goals regarding fat and added sugar consumption. Specifically, we reference the “2020-2025 Dietary Guidelines for Americans, which recommend that saturated fat and added sugars are limited to <10% of total daily energy intake” in the introduction (Lines 48-51) and in the discussion section (Lines 260-261). We also now refer to the Academy of Nutrition and Dietetics Evidence-based Nutrition Practice Guideline regarding interventions on saturated fat intake, stating that they should include consideration of individual preferences etc. (Lines 295-298).
Furthermore, it was suggested that the authors review the references used, as there are generic sentences that could be supported by a single reference but are instead based on up to five references.
Response: We have reviewed our references where noted (Lines 36-42) and reduced the references.
This manuscript is a resubmission of an earlier submission. The following is a list of the peer review reports and author responses from that submission.
Round 1
Reviewer 1 Report
Comments and Suggestions for Authors
I agree with the authors that anything that can be done to streamline and improve methods of dietary intake are worthwhile endeavors. However, I do not fully agree that narrowing down a list to 95 SF/AS foods is really simplifying anything. The research is sound, but it is hard to understand the benefit.
Additionally, the anthropometric data of the study population and groups are not presented anywhere that I can find. I realize that the majority of your findings are expressed as percent of total intake, which helps with some potential inconsistencies in measuring dietary intake, but the characterization of the study population is still important.
Reviewer 2 Report
Comments and Suggestions for Authors
The paper does not adequately justify the specific focus on saturated fats and added sugars in the US adult population, given the extensive existing research in this area. The rationale for another study replicating similar methodologies and results without introducing new hypotheses or exploring under-researched areas appears weak. This may further emphasise the lack of novelty and significant contribution to the field, which is essential for publication in a scientific journal. Please see similar paper: https://www.jandonline.org/article/S2212-2672(21)01121-7/abstract https://www.mdpi.com/2072-6643/15/2/265
Check the file for plagiarism
Lack of originality in methodology: The methods used to analyse dietary data, particularly the reliance on NHANES data and standard dietary assessment techniques, do not provide a new perspective or innovative approach that distinguishes this work from previous studies.
Inadequate statistical analysis: The statistical methods used to analyse the data are elementary and do not apply advanced statistical techniques that could have provided insights into the data, such as interactions between different dietary components and demographic variables.
Superficial discussion of results: The discussion section lacks depth in the interpretation of the results, failing to effectively link the results to broader nutritional science or public health implications. This section does not add value to what is already available in the literature cited.
Poor integration of literature: The literature review in the introduction and discussion sections seems superficial and does not adequately integrate existing research to build a compelling case for the relevance and need for the study. This lack of integration fails to effectively position the paper within the ongoing scientific conversation.
